# Plasma and breast milk adipokines in women across the first year postpartum and their association with maternal depressive symptoms and infant neurodevelopment: Protocol for the APPLE prospective cohort study

Fernanda Rebelo[1]*, Cintia Oliveira de Moura[2], Layla Galvão Ranquine[3], Thaisa de Mattos Teixeira[4], Mariana Terra Nunes Ribas[3], Raquel Santiago Vitorino[4], Andrea Dunshee de Abranches[1], Roozemeria Pereira Costa[1], José Augusto Alves de Britto[5], Daniele Marano[1], Fernanda Valente Mendes Soares[1], Maria de Fátima Junqueira-Marinho[1], Carlos Augusto Moreira de Sousa[6], Ana Beatriz Franco-Sena[7], Antônio Egídio Nardi[8], Tatiana El-Bacha[3], Maria Elisabeth Lopes Moreira[1]

1 Instituto Nacional de Saúde da Mulher, Unidade de Pesquisa Clínica, da Criança e do Adolescente Fernandes Figueira, Fundação Oswaldo Cruz, Rio de Janeiro, RJ, Brazil, 2 Instituto Nacional de Saúde da Mulher, Programa de Pós-graduação em Pesquisa Aplicada à Saúde da Criança e da Mulher, da Criança e do Adolescente Fernandes Figueira, Fundação Oswaldo Cruz, Rio de Janeiro, RJ, Brazil, 3 Instituto de Nutrição Josué de Castro, LeBioME-Bioactives, Mitochondrial and Placental Metabolism Core, Universidade Federal do Rio de Janeiro, Rio de Janeiro, RJ, Brazil, 4 Instituto Nacional de Saúde da Mulher, Programa de Pós-graduação em Saúde da Criança e da Mulher, da Criança e do Adolescente Fernandes Figueira, Fundação Oswaldo Cruz, Rio de Janeiro, RJ, Brazil, 5 Instituto Nacional de Saúde da Mulher, Área da Pediatria–Unidade Ambulatorial, da Criança e do Adolescente Fernandes Figueira, Fundação Oswaldo Cruz, Rio de Janeiro, RJ, Brazil, 6 Faculdade de Ciências Médicas, Departamento de Tecnologias da Informação e Educação em Saúde, Universidade do Estado do Rio de Janeiro, Rio de Janeiro, RJ, Brazil, 7 Faculdade de Nutrição Emília de Jesus Ferreiro, Departamento de Nutrição Social, Universidade Federal Fluminense, Niterói, RJ, Brazil, 8 Instituto de Psiquiatria, Universidade Federal do Rio de Janeiro, Rio de Janeiro, RJ, Brazil

☯ These authors contributed equally to this work.
* frebelos@gmail.com

## Abstract

### Introduction

Adiponectin and leptin play important roles in the central nervous system. During the post-partum period, there is a need for a better understanding of the relationship between these cytokines and the neurological development of the infant, as well as their influence on preventing maternal depressive symptoms.

### Objectives

To assess the correlation between adiponectin and leptin in maternal plasma and breast milk and their association with: infant neurodevelopment at 6 and 12 months of age; and maternal mental health over the first year postpartum.

**Data Availability Statement:** No datasets were generated or analysed during the current study. All relevant data from this study will be made available upon study completion. This article does not report any data, so the data availability policy is not applicable. Upon completion of the APPLE project, the final database containing de-identified research data will be deposited in Arca Dados (https://arcadados.fiocruz.br), the official repository of Fundação Oswaldo Cruz.

**Funding:** The project is being funded by the Research Support Program (APQ-1) of the Rio de Janeiro State Research Support Foundation (FAPERJ Proc. No. 211.557/2021). Participant reimbursement is being provided with support from the Research Productivity Fellowship from the National Council for Scientific and Technological Development (CNPq) under the name of AEN. COM, LGR and TMT receive scholarships from Coordination for the Improvement of Higher Education Personnel (CAPES). MTNR receives a scholarship from CNPq. The funders had no role in study design, data collection and analysis, decision to publish, or preparation of the manuscript.

**Competing interests:** The authors have declared that no competing interests exist.

## Methods

Prospective cohort study with four follow-up. Mothers and their newborns are recruited within the first 15 days postpartum (baseline). Follow-up visits occur at 2, 6, and 12 months postpartum. Visits include blood and breast milk collection, application of the Edinburgh Postnatal Depression Scale and Beck Depression Inventory to assess maternal mental health, application of the Bayley-III scale for infant developmental assessment, maternal and infant anthropometry and body composition, evaluation of reproductive history, mother-infant bonding, breastfeeding, consumption of ultra-processed foods, sleep quality, and socio-economic and demographic data.

## Results

The research received funds in August 2022, and participant recruitment began in September 2022. The sample size will consist of 95 mother-child pairs. As of September 2023, 68 participants have been recruited.

## Conclusion

The project will provide insights into the association between adiponectin and leptin with postpartum depression and infant neurodevelopment, ultimately promoting improved care and quality of life for these groups. Additionally, it will provide data on the type of delivery, infant physical growth, maternal and infant body composition changes, sleep quality, consumption of ultra-processed foods, and maternal metabolic health, including vitamin D metabolites, oxidized polyunsaturated fatty acid metabolites, phospholipid species and triacylglycerols, which are of significant relevance to public health and, when interconnected, may yield important results and contribute to the existing literature.

## Trial registration

**Name of the registry:** Brazilian Clinical Trials Registry (ReBec).

**Registration number:** RBR-9hcby8c.

## Introduction

Human milk contains various nutrients, cytokines, peptides, enzymes, cells, immunoglobulins, proteins, and steroids capable of meeting the needs of the infant [1–3]. Some of these compounds are synthesized by the mammary glands, while others are taken from the maternal plasma. Maternal nutritional status, supplement usage, and dietary intake are some of the factors that can modify the plasma concentrations of these compounds and, consequently, the composition of the milk [4–7].

Among the cytokines already identified in breast milk are adiponectin and leptin, also known as biomarkers related to obesity or adipokines, as they are secreted by adipose cells [8]. It is believed that these cytokines found in breast milk have hormonal activities in various tissues of the neonate, especially when their own endocrine system is not fully functional, playing roles in energy metabolism and the regulation of body composition [9].

Additionally, studies suggest that adipokines may have activities in the brain, such as stimulating the genesis and neuronal excitability of the hippocampus, contributing to functions like

learning and memory [10–14]. In recent years, the association between adipokines and cognitive function has been studied in other population groups, such as adults and the elderly [15–17]. More recently, some studies have provided data on the pediatric population.

In a cross-sectional study involving 50 participants conducted in Brazil, plasma concentrations of leptin were found to be inversely associated with cognitive development in children aged 6 to 24 months [18]. Using data from two cohorts of pregnant women in the United States and Canada, Li et al. (2019) found an association between adiponectin in umbilical cord blood and the intelligence quotient and memory in children at 3, 5, and 8 years of age [19]. However, to date, no studies have been found that investigate the association of maternal adipokines with the neurodevelopment of infants during the first year of life.

The period during which brain growth reaches its highest speed and exhibits the greatest plasticity occurs in the last trimester of pregnancy and the first two years of life. Therefore, the first one thousand days of life, from conception to 2 years of age, are recognized as crucial in the development trajectory, determining future health status, and being linked to increased cognitive and learning abilities throughout adulthood [20–22]. Hence, the identification of factors that may be associated with neurodevelopment in the first year of life, considered a window of opportunity for interventions, is of great significance in contributing to the health and quality of life of the population.

Regarding maternal outcomes associated with adipokine concentrations in postpartum women, further clarification is needed, particularly concerning mental health. The literature provides evidence that individuals with conditions such as depression and anxiety may have altered plasma concentrations of adipokines [23–25]. However, data regarding the relationship between adipokines and mental health during the postpartum period are limited and inconclusive [26].

Postpartum depression is particularly problematic because it not only impacts the health of the woman but also has a negative effect on the child's health and complicates the use of medication due to the transmission of the drug to the infant through breast milk [27–30]. Mothers with postpartum depression often experience compromised physical health and engage in high-risk behaviors, such as alcohol and substance abuse. These women are less likely to adequately care for their own needs, as well as those of their children, and they may exhibit greater resistance to seeking and receiving postnatal care or adhering to medical prescriptions [31]. Moreover, suicide, with depression as its primary risk factor, stands as the foremost cause of maternal mortality in developed nations [32].

The role of adipokines in the central nervous system has been the focus of numerous studies over the last decade. In vitro studies demonstrate the role of adiponectin and leptin in stimulating neurogenesis, regulating the proliferation of hippocampal cells in a time- and dose-dependent manner [33,34]. Hence, the promotion of neurogenesis is a potential mechanism through which these adipokines can act on both stimulating infant neurodevelopment and inhibiting postpartum depressive symptoms. Additionally, adiponectin may indirectly influence depression through its actions on the hypothalamic-pituitary-adrenocortical axis (suppression of TNFα) and/or thyroid-mediated effects [13].

Given the scarcity of studies on this subject, the current project stands out for its potential innovation and ability to generate evidence that can assist in understanding and preventing developmental issues in childhood and throughout the life cycle. It also aims to contribute to the search for new strategies to prevent and treat postpartum depression.

In this way, the project seeks to clarify the relationship between adiponectin and leptin in maternal plasma and breast milk, maternal mental health, and infant neurodevelopment. This is based on the hypothesis that plasma concentrations of adiponectin and leptin are associated with the mental health of the breastfeeding mother and are correlated with the concentrations

of these adipokines in breast milk. In turn, the consumption of milk with higher concentrations of adiponectin and lower leptin is expected to lead to greater neural stimulation in the infant, promoting better neurodevelopment.

In summary, the research questions to be responded to are: What is the relationship between maternal mental health and the concentrations of adiponectin and leptin in maternal plasma? How are the concentrations of adiponectin and leptin in maternal plasma related to their levels in breast milk? Does the consumption of breast milk with higher adiponectin and lower leptin concentrations result in improved neurodevelopmental outcomes in infants? What role do adiponectin and leptin play in the neurodevelopment of infants, particularly in the context of maternal mental health? These questions aim to dissect the intricate connections between maternal health, breast milk composition, and infant neurodevelopment, offering potential pathways for improving child health outcomes through targeted maternal support.

## Methods

To facilitate communication with research participants and reference to the project within the healthcare unit where the project takes place, the research title, "Adipokines in the plasma and milk of breastfeeding mothers in the first year postpartum: association with depressive symptoms and infant neurodevelopment," has been abbreviated to the acronym APPLE.

A prospective cohort study is being conducted at the National Institute of Women's, Children's, and Adolescents' Health Fernandes Figueira (IFF), a healthcare unit affiliated with the Oswaldo Cruz Foundation (Fiocruz), located in Rio de Janeiro, Brazil. The IFF houses a maternity ward where approximately 1,000 deliveries occur annually. The cohort comprises four follow-ups: immediate postpartum (up to 15 days after delivery), the 2nd month postpartum, the 6th month postpartum, and the 12th month postpartum. Adult breastfeeding mothers and their children who are within the first 15 days postpartum are invited to participate in the study.

The recruitment of participants is being carried out through two approaches: (1) Project promotion for postpartum women who have given birth at IFF, followed by an active search for eligible patients who return for pediatric care at the Human Milk Bank (HMB) at IFF; (2) Promotion to personal contacts of the project team is conducted through messaging apps and social media platforms and interested individuals reach out via email or phone for eligibility assessment and scheduling of the initial evaluation.

The follow-up takes place at the HMB (milk collection), the pediatric service (infant development assessment), and the Nutrition and Metabolism Laboratory (body composition assessment, blood collection, and questionnaire administration). The specifics of the variables obtained at each follow-up wave of the study can be seen in **Fig 1**.

To be included, breastfeeding mothers should be within 15 days postpartum, aged between 20 and 45 years, practicing exclusive breastfeeding (offering only breast milk, without tea or water, in the 5 days preceding entry into the study), residing in the city of Rio de Janeiro, having proficiency in the Portuguese language, and their newborn should have had an Apgar score of $\geq 7$ at the fifth minute.

Breastfeeding mothers meeting the following criteria are ineligible for the study: those with chronic non-communicable diseases such as hypertension and diabetes (with the exception of obesity); individuals with infectious diseases like HIV and syphilis; those using antidepressants or other psychiatric medications; those who experienced a twin pregnancy; those with positive serology for Syphilis, Rubella, Toxoplasmosis, Cytomegalovirus, or Zika during pregnancy; and those who engaged in gestational surrogacy or temporary uterine cession ("surrogate mother"). Furthermore, puerperal women are ineligible if their newborns were preterm,

| | STUDY PERIOD | | | |
|---|---|---|---|---|
| | Enrolment | Follow-up | | |
| **TIMEPOINT** | $t_0$ | $t_1$ | $t_2$ | $t_3$ |
| **ENROLMENT:** | | | | |
| **Eligibility screen** | X | | | |
| **Informed consent** | X | | | |
| **ASSESSMENTS:** | | | | |
| *Maternal blood samples* | X | | X | X |
| *Maternal human milk samples* | X | X | X | X |
| *Maternal depressive symptoms (EPDS)* | X | X | X | X |
| *Infant neurodevelopment (BSID-III)* | | | X | X |
| *Socioeconomic, demographic, lifestyle, and reproductive history variables* | X | | | |
| *Maternal anthropometry and body composition* | X | X | X | X |
| *Infant anthropometry* | X | X | X | X |
| *Infant body composition* | X | X | | |
| *Breastfeeding practices* | | X | X | X |
| *Maternal dietary intake of ultra-processed foods (NOVA score)* | X | X | X | X |
| *Sleep quality (PSQI-BR)* | X | X | X | X |
| *Infant dietary intake* | | X | X | X |
| *Mother-infant emotional bond (PBQ)* | | X | X | X |
| *Maternal depression diagnosis (BDI-II)* | | | X | X |
| *Maternal dietary intake (Nova24h)* | | | | X |
| *Social support (MOS)* | | | | X |

**Fig 1. Schedule of enrolment and assessments for the APPLE prospective cohort study.**

exhibited any pathological conditions, had inappropriate weight for their gestational age, were diagnosed with conditions or malformations known to cause motor and cognitive deficits, or were admitted to a neonatal intensive care unit.

Regarding postpartum mothers who gave birth at IFF, their medical records are reviewed by trained professionals to assess eligibility for the study three times a week on alternate days, typically on Mondays, Wednesdays, and Fridays. On Fridays, assessments are conducted in the morning and early afternoon to minimize potential losses, considering that discharges from the maternity ward occur 2 or 3 days after delivery, depending on the type of delivery. Postpartum mothers identified as eligible through the medical record analysis are approached in the maternity unit for the initial project presentation. They receive a flyer with general information and contact details for the project, including phone and email.

In this scenario, the first meeting takes place on the pediatric evaluation date, which is a scheduled appointment at the Human Milk Bank (BLH) of IFF, at the time of maternity discharge, by the pediatric team. Between one and two days before the scheduled date, the project team contacts the postpartum mother to confirm her attendance at the appointment or reschedule it, aiming to prevent any losses due to forgetfulness or an inability to attend the originally scheduled date.

For postpartum mothers outside the IFF, promotion is being conducted among the team's contacts through messaging apps and social media. Our outreach encourages interested pregnant individuals to register via an online form to assess eligibility and the expected delivery date. Closer to the expected delivery date, we contact the pregnant individuals to confirm their interest and schedule their participation. A new eligibility check is done by phone one day before the scheduled date to prevent unnecessary travel. If a woman has already given birth when she receives the promotion, we encourage her to contact a team member directly by phone, where eligibility criteria are assessed, and the evaluation is scheduled.

At the end of each assessment, postpartum mothers are scheduled for the next follow-up. To minimize follow-up losses, all participants are contacted via messaging apps or phone for confirmation or rescheduling. Another strategy to increase adherence includes sending a monthly image congratulating them on another month of their baby's life on the baby's birth date. This image also contains general information about the baby's health during that period, such as developmental milestones, feeding, and vaccinations.

The participants started to be recruited in September 2022. Recruitment was open until the desired sample size is achieved (n = 117), which happened in July 2024. The follow-up will continue until all participants reach the one-year postpartum mark, projected to occur in July 2025.

## Variables of the study

**Infant neurodevelopment (outcome 1).** Neurodevelopment is assessed using the Bayley Scales of Infant Development (BSID-III) at the 6th and 12th month after childbirth (with a margin of 15 days either way from the exact date). The scale is administered by a trained professional (psychologist) from Pearson. This instrument is designed for assessing children aged 1–42 months and comprises the following scales: (i) Cognitive Scale, evaluating sensory-motor development, exploration and manipulation, concept formation, and memory; (ii) Language Scale, consisting of items for receptive and expressive communication, forming two distinct sub-tests; (iii) Motor Scale, assessing overall motor skills and fine motor abilities; (iv) Socio-Emotional Scale, identifying key milestones in social and emotional development at specific ages; (v) Adaptive Behavior Scale, accessing the child's functional daily life skills, considering communication, home life, health, safety, and leisure.

Additionally, members of the project team assess the developmental milestones recorded in the Child's Health Record. This assessment is routinely conducted in the pediatric clinic, and its validity, as compared to the results of the BSID-III (gold standard), will provide another valuable piece of data generated by the current study.

**Maternal mental health (outcome 2).** Maternal mental health is monitored in all waves of follow-up: up to 15 days, 2, 6, and 12 months postpartum. During the first appointment, data are collected regarding the postpartum mother's history of depression, as well as her family history of depression. A validated version of the Edinburgh Postnatal Depression Scale (EPDS) is employed. The EPDS was developed by Cox et al. [35] and translated and validated into Portuguese by Santos et al. [36] and consists of a self-report instrument composed of 10 statements, with scored options from 0 to 3 based on the presence and intensity of depressive symptoms.

Many studies cite and use the EPDS because it is a questionnaire that is easy to administer and understand for women, and it has proven to be highly effective in identifying postpartum depression [37–40]. At baseline, the EPDS will be administered by trained interviewers (health professionals or undergraduate students in the healthcare field). In the subsequent follow-ups, participants will receive the EPDS online, via email and/or messaging app, seven days before the in-person meeting, allowing them to self-complete the questionnaire. If a participant does not respond to the online questionnaire, the scale will be administered by trained interviewers on the day of the in-person meeting.

Additionally, depression is assessed using the Beck Depression Inventory-II (BDI-II) at 6 and 12 months postpartum. This is because the validation of the Brazilian version of the EPDS was only conducted with women up to 3 months postpartum. The BDI-II is a widely used instrument worldwide and has been validated for the Brazilian population [41]. The application will be conducted by a trained psychologist.

**Maternal blood and breast milk samples (main exposure).** Human milk (a minimum of 10 ml and a maximum of 20 ml) is obtained using an electric breast pump. These samples are stored and transported under refrigeration within a maximum of 2 hours to the location for analysis. The samples are homogenized using a vortex (1 minute), divided into aliquots, and stored at -30°C for a maximum of 24 hours, after which they are moved to final storage at -80°C, where they remain until processing and analysis. Samples are collected at every follow-up visit or until the breastfeeding mother no longer has milk.

Blood samples are collected at visits T0, T2, and T3 in collection tubes containing EDTA. The samples are centrifuged (3500 rpm/15 minutes), and the plasma is separated into aliquots and stored in a freezer at -30°C for a maximum of 24 hours, after which they are transferred to final storage at -80°C, where they remain until processing and analysis. Commercial ELISA kits are used to determine the concentration of adiponectin and leptin in both breast milk and plasma.

In addition to adipocytokines, inflammatory biomarkers, including non-enzymatic oxidized metabolites of polyunsaturated fatty acids and species of phospholipids and triacylglycerols, as well as vitamin D metabolites, will be investigated. The oxidized metabolites and vitamin D metabolites will be analyzed through targeted lipidomics using a QTrap mass spectrometry platform (AB Sciex QTRAP 5500) [42], and lipid species through high-resolution semi-targeted lipidomics on an Orbitrap platform (Q-Exactive, Thermo-Finnigan) [43].

**Socioeconomic, demographic, lifestyle, and reproductive history variables.** A structured questionnaire will be applied to determine maternal socioeconomic and demographic characteristics, as well as lifestyle and reproductive data that are part of the mother's history: age, marital status, number of children, occupation, level of education, place of residence, type of housing, smoking, alcohol consumption, social support, newborn's sex, number of births,

date of the last birth, type of delivery (specifying the reason for the surgical delivery, when applicable), and duration of pregnancy.

Social support is estimated through two simple questions at T1 and T2: "Do you share baby care with someone?" (response options: yes, no) and "How often do you have someone to help you if you are very tired or sick?" (response options: never, rarely, sometimes, almost always, always). At T3, social support is measured using the Portuguese version of the social support questionnaire from the Medical Outcomes Study [44]. This instrument assesses the perceived availability of support when needed and consists of four dimensions: material, affective, positive social interaction, and emotional/informational. The instrument comprises 19 questions with responses ranging from 1 (never) to 5 (always) points. Therefore, social support is measured by summing the responses, and the higher the score, the greater the estimated social support.

**Maternal and infant anthropometry.** At the first appointment, data on the woman's pre-pregnancy weight will be collected. Preference will be given to the weight measured up to the 13th week of pregnancy recorded in the pregnant woman's follow-up booklet. If this record is nonexistent, the reported weight will be recorded. For the newborn, the following data contained in the Child Health Handbook will be collected: birth weight, length at birth, head circumference at birth, and Apgar scores at 1 and 5 minutes.

For the assessment of the body composition of postpartum women (in all follow-ups) and infants (T0 and T1), the air displacement plethysmography method will be used. This method evaluates body volume and density by measuring the volume of air displaced by the body inside a closed chamber. It is a relatively recent, fast, safe, and comfortable method [45]. This technique is based on Boyle's law, which relies on the inverse relationship between pressure and volume to determine body composition. Once this volume is defined, it is possible to apply the principles of densitometry to determine body composition by calculating body density [46]. The plethysmograph, also known as the BOD POD for adults or PEA POD for infants, is connected to a computer that calculates variations in air volume and pressure inside the empty and occupied chamber. This equipment makes adjustments for lung variables required in estimating body volume. The equipment also provides the weight of the woman and the child during each assessment. Women undergo the assessment wearing tight-fitting clothing, such as a top and shorts or a swimsuit, and they use a swim cap to secure their hair. Infants are assessed without clothing and wear a stocking cap on their heads, regardless of whether they have hair or not.

Maternal height will be directly assessed using a wall-mounted compact stadiometer (WISO®) with a precision of one millimeter during the first follow-up wave.

Starting from T2, the child's weight will be measured using a digital pediatric scale with a maximum capacity of 20kg or on the scale integrated with the PEA POD. The child's length will be measured using an infantometer, with the child lying down and their head supported on the fixed part of the instrument. The movable part will be appropriately positioned under the child's feet. Measurements will be taken in duplicate, and the average of the measurements will be used. Additionally, the child's fronto-occipital head circumference will be measured using a non-stretchable measuring tape, corresponding to the maximum head circumference.

**Breastfeeding and infant dietary intake.** The frequency of breastfeeding will be estimated in all follow-ups using a structured questionnaire adapted from the II Breastfeeding Prevalence Survey in Brazilian Capitals and the Federal District, conducted by the Ministry of Health [47]. The questionnaire also assesses the introduction of solid foods by inquiring about the consumption of porridge, fruits, savory foods, beans, meats, vegetables, greens, and ultra-processed foods.

**Maternal dietary intake.** The dietary intake of ultra-processed foods will be assessed in all follow-up waves using the NOVA score [48]. This is a self-administered questionnaire that

takes around three minutes to complete and has shown good performance in estimating the consumption of ultra-processed foods in the Brazilian diet. The questionnaire will be sent electronically (via email or WhatsApp) one week before the in-person consultations (except for the first consultation). For women who do not respond to the electronic questionnaire, it will be administered during the in-person consultation.

The overall dietary intake of postpartum women is assessed in the third follow-up (T3) using a 24-hour dietary recall proposed by Neri et al. (2023) (Nova24h, Portuguese version) [49]. It's a self-administered web-based instrument designed to categorize food intake according to the Nova groups. The Nova24h will be applied on two non-consecutive days to estimate habitual consumption. The first dietary recall will be sent via email prior to the T3 consultation. The second will be completed on the day of the T3 consultation. Dietary data will be entered into the Nutrabem Pro®14 software to obtain energy, macro, and micronutrient values.

**Sleep quality.** The quality of sleep will be assessed using the Pittsburgh Sleep Quality Index, translated and validated for the Brazilian population (PSQI-BR) [50]. In the first wave of follow-up (up to 15 days postpartum), the questions will relate to the gestational period. In the subsequent waves, women will be asked to respond based on the current period. The questionnaire will be sent electronically (via email or WhatsApp) one week before the in-person appointments (except for the first appointment). For women who do not respond to the electronic questionnaire, it will be administered during the in-person appointment.

**Mother-infant emotional bond.** The mother-infant emotional bond will be assessed in the second and third follow-up waves (T1 and T2) through the translated version of the Postpartum Bonding Questionnaire (PBQ) [51]. The PBQ was originally developed by Brockington & Wilson and consists of 25 items divided into four factors: overall emotional bonding, rejection and anger towards the baby, anxiety directed towards the baby, and the risk of abuse [52]. Participants rate their level of agreement with the statements on Likert scales ranging from 0 (always) to 5 (never). The questionnaire will be sent electronically (via email or WhatsApp) one week before the in-person consultations. For women who do not respond to the electronic questionnaire, it will be administered during the face-to-face consultation.

**Participant support.** The occurrence of depressive symptoms in women is the critical outcome that is expected to be more recurrent during the project's follow-up. Women classified with depressive symptoms by the EPDS scale ($\geq$ 13 points) or diagnosed with depression by the BDI-II are referred for care at the Mental Health Coordination of IFF (COJ) if they are enrolled at IFF. Participants who are not enrolled at IFF are referred for care at the outpatient clinic of the Institute of Psychiatry of the Federal University of Rio de Janeiro (IPUB). Naturalistic medical and psychological treatment will be provided according to the postpartum period and the patient's symptoms.

In the case of other critical outcomes, the project team is responsible for ensuring that participants receive appropriate care. It is worth noting that the team is multidisciplinary, consisting of professionals from the fields of pediatrics, psychiatry, neonatology, psychology, nutrition, and nursing.

The total amount spent on transportation and meals related to participation in the project is reimbursed to the participants.

## Sample size

To assess differences in adipocytokines in human milk according to infant development, the reference was the global score of cognitive development in the communication domain. A previous study assumed that the standard deviation of the score is 13.6 for children of 6 months

[53]. We will consider a difference of 3 points significant, and we aim for a significance level of 5% to detect differences if they exist.

Applying the formula for determining the sample size based on an estimate of the population mean, $n = [(Z_{\alpha/2}.\sigma)/e]^2$, where $Z_{\alpha/2} = 1.96$, e = 3 and σ = 13.6 (standard deviation), we obtain N = 79 [54]. If we estimate a follow-up loss of approximately 35%, the sample size (N) should be 117 mother-child pairs.

## Data entry and statistical analysis

The collected data is being entered into the database using data entry masks specifically developed for the study in the RedCap software. Data storage and backup are provided by application and database servers located in the secure room of COGETIC/Fiocruz, with security levels in compliance with current legislation. A full backup is performed daily, and log backups are done hourly every day from 08:00 to 18:00.

The statistical analyses will be conducted using the Stata statistical software package [55]. Initially, data will be evaluated through a dispersion analysis. The normality of the data will be assessed using the Shapiro-Wilk test, and non-normally distributed variables will be subjected to a logarithmic transformation for subsequent analysis using parametric tests. Proportions of categorical variables will be compared using the chi-squared test, and differences between means of continuous variables will be tested with the Student's t-test or ANOVA. Pearson's correlation coefficient will be used to analyze the correlation between continuous variables, such as the correlation between adiponectin and leptin in plasma and milk.

The main variables (adiponectin and leptin concentrations, development scores, and depressive symptoms) will be treated as continuous. To assess the association between these variables, a longitudinal mixed-effects model (LME) will be used, adjusted for confounding factors. LME is a model capable of capturing changes within and between individuals, accommodating time-dependent and independent covariates, accounting for the correlation between repeated measurements in the same individual, and allowing for unbalanced time intervals. The total variance will be estimated by two components: the first measures the variance between individuals at the same time, and the second component measures the within-individual variance between each time point during follow-up. Postpartum time will be included as a fixed and random effect variable. If the response is described by a nonlinear function, a quadratic term will be included in the model.

To determine the model's adjustment variables, a Directed Acyclic Graph (DAG) will be constructed using the online software DAGitty [56]. The purpose of the DAG is to improve our understanding of the association between exposure and outcome, along with all potential confounding factors. This approach allows us to identify the minimally sufficient adjustment required to estimate the direct effect of plasma adiponectin concentrations on infant neurodevelopment.

## Ethical issues

The project complies with Brazilian Resolutions No. 466 of December 12, 2012, and No. 510 of April 7, 2016. Participation in the research is contingent upon signing a two-copy informed consent form, which will be obtained freely and voluntarily after all relevant clarifications about the present study have been made. The study was approved by the Research Ethics Committee of IFF (CAAE: 26971319.1.0000.5269, opinion: 6.095.803, version 13). Any modifications to the protocol will be submitted for ethical review before implementation.

The protocol was registered in the Brazilian Clinical Trials Registry (ReBec) on December 19th, 2023 (registration number: RBR-9hcby8c; access: https://ensaiosclinicos.gov.br/rg/RBR-

9hcby8c). The study was not registered prior to the enrolment of participants, given its observational nature, where registration is not obligatory.

The original and translated protocols are available as (S1 and S2 Protocols). As a request from the editorial board of PLOS ONE, we are also presenting the SPIRIT Checklist as Supporting Information (S1 Checklist).

## Discussion

This project aims to, first and foremost, advance the understanding of the relationships between plasma and breast milk adipokines, infant neurodevelopment, and maternal mental health. Additionally, the project will provide data on the type of delivery, infant physical growth and weight variation, sleep quality, social support, maternal consumption of ultra-processed foods, and maternal metabolic health, including vitamin D metabolites and fatty acids and their oxidized products, in the first year postpartum. These data, of great relevance to public health, when interrelated, could yield important results and contribute to the existing literature.

Some in vitro studies already suggest that adipokines have roles in neuronal regulation. It appears that this will be the first epidemiological study to provide evidence on the association of maternal adipokines with infant cognitive development and maternal mental health in the first year postpartum. At this point, an observational study that demonstrates these potential associations will generate evidence, raise hypotheses, and inspire further studies on this topic.

The results are expected to serve as an incentive for breastfeeding, as well as to aid in understanding and preventing developmental issues in childhood and throughout the life cycle. In the future, investigating factors associated with adipokine concentrations in breast milk and interventions that can modulate these concentrations may lead to new recommendations for postpartum women, impacting children's learning capacity and, consequently, improving their social, emotional, and cognitive well-being.

Additionally, considering the need for more evidence on the association between adipokines and depression, the lack of studies on this topic in postpartum women, and the impact of depression on maternal and child health, assessing the association between adipokines and maternal mental health is a pathway to explore new strategies for preventing and treating postpartum depression. If the hypothesis is confirmed, clinicians could consider monitoring adiponectin and leptin levels in maternal plasma as potential biomarkers for maternal mental health and infant neurodevelopmental outcomes. Furthermore, this research could inform guidelines for supporting breastfeeding mothers, emphasizing the importance of maternal nutrition and mental well-being for optimizing the neurodevelopment of the infant. Ultimately, this could lead to more personalized approaches in maternal and infant healthcare, where interventions are tailored based on the specific needs of both the mother and child.

The main limitations of this study are considered to be the potential losses of follow-up, which are inherent to longitudinal studies, and the relatively small sample size. While the sample size is sufficient for the primary objectives of the project (as indicated by the sample size calculation), some specific objectives may not be fully addressed, thereby increasing the risk of type II errors.

## Supporting information

**S1 Checklist. SPIRIT checklist.**
(DOC)

**S1 Protocol. Original protocol submitted to the Ethics committee (Portuguese).**
(PDF)

**S2 Protocol. Original protocol submitted to the Ethics committee (translated to English).**
(PDF)

## Acknowledgments

We would like to thank the entire team involved in data collection, our colleagues at IFF for sharing their spaces, equipment, and experiences, and the puerperal women and their children who have agreed to participate in this research.

## Author Contributions

**Conceptualization:** Fernanda Rebelo, Cintia Oliveira de Moura, Layla Galvão Ranquine, Thaisa de Mattos Teixeira, José Augusto Alves de Britto, Daniele Marano, Fernanda Valente Mendes Soares, Maria de Fátima Junqueira-Marinho, Carlos Augusto Moreira de Sousa, Ana Beatriz Franco-Sena, Antônio Egídio Nardi, Tatiana El-Bacha, Maria Elisabeth Lopes Moreira.

**Formal analysis:** Fernanda Rebelo, Carlos Augusto Moreira de Sousa.

**Funding acquisition:** Fernanda Rebelo, Antônio Egídio Nardi.

**Investigation:** Fernanda Rebelo, Cintia Oliveira de Moura, Layla Galvão Ranquine, Thaisa de Mattos Teixeira, Mariana Terra Nunes Ribas, Raquel Santiago Vitorino, Andrea Dunshee de Abranches, Roozemeria Pereira Costa, José Augusto Alves de Britto, Daniele Marano, Fernanda Valente Mendes Soares, Maria de Fátima Junqueira-Marinho, Ana Beatriz Franco-Sena, Tatiana El-Bacha, Maria Elisabeth Lopes Moreira.

**Methodology:** Fernanda Rebelo, Cintia Oliveira de Moura, Layla Galvão Ranquine, Thaisa de Mattos Teixeira, Mariana Terra Nunes Ribas, Raquel Santiago Vitorino, Andrea Dunshee de Abranches, Roozemeria Pereira Costa, José Augusto Alves de Britto, Daniele Marano, Fernanda Valente Mendes Soares, Maria de Fátima Junqueira-Marinho, Carlos Augusto Moreira de Sousa, Ana Beatriz Franco-Sena, Antônio Egídio Nardi, Tatiana El-Bacha, Maria Elisabeth Lopes Moreira.

**Project administration:** Fernanda Rebelo, Cintia Oliveira de Moura, Layla Galvão Ranquine, Thaisa de Mattos Teixeira.

**Resources:** Antônio Egídio Nardi, Tatiana El-Bacha, Maria Elisabeth Lopes Moreira.

**Supervision:** Fernanda Rebelo, Cintia Oliveira de Moura, Layla Galvão Ranquine, Thaisa de Mattos Teixeira.

**Writing – original draft:** Fernanda Rebelo.

**Writing – review & editing:** Cintia Oliveira de Moura, Layla Galvão Ranquine, Thaisa de Mattos Teixeira, Mariana Terra Nunes Ribas, Raquel Santiago Vitorino, Andrea Dunshee de Abranches, Roozemeria Pereira Costa, José Augusto Alves de Britto, Daniele Marano, Fernanda Valente Mendes Soares, Maria de Fátima Junqueira-Marinho, Carlos Augusto Moreira de Sousa, Ana Beatriz Franco-Sena, Antônio Egídio Nardi, Tatiana El-Bacha, Maria Elisabeth Lopes Moreira.

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
