## [Decision Letter · Decision Letter 0]

14 Aug 2024

PONE-D-23-34502Plasma and breast milk adipokines in women across the first year postpartum and their association with maternal depressive symptoms and infant neurodevelopment: protocol for the APPLE prospective cohort studyPLOS ONE

Dear Dr. Rebelo,

Thank you for submitting your manuscript to PLOS ONE. After careful consideration, we feel that it has merit but does not fully meet PLOS ONE’s publication criteria as it currently stands. Therefore, we invite you to submit a revised version of the manuscript that addresses the points raised during the review process.

We look forward to receiving your revised manuscript.

Kind regards,

Annisa Dewi Nugrahani

Academic Editor

PLOS ONE

2. We note that you have selected “Clinical Trial” as your article type. PLOS ONE requires that all clinical trials are registered in an appropriate registry (the WHO list of approved registries is at https://www.who.int/clinical-trials-registry-platform/network/primary-registries" https://www.who.int/clinical-trials-registry-platform/network/primary-registries and more information on trial registration is at http://www.icmje.org/about-icmje/faqs/clinical-trials-registration/). Please state the name of the registry and the registration number (e.g. ISRCTN or ClinicalTrials.gov) in the submission data and on the title page of your manuscript. a) Please provide the complete date range for participant recruitment and follow-up in the methods section of your manuscript. b) If you have not yet registered your trial in an appropriate registry, we now require you to do so and will need confirmation of the trial registry number before we can pass your paper to the next stage of review. Please include in the Methods section of your paper your reasons for not registering this study before enrolment of participants started. Please confirm that all related trials are registered by stating: “The authors confirm that all ongoing and related trials for this drug/intervention are registered”. Please see http://journals.plos.org/plosone/s/submission-guidelines#loc-clinical-trials for our policies on clinical trials. 

 [The project is being funded by the Research Support Program (APQ-1) of the Rio de Janeiro State Research Support Foundation (FAPERJ Proc. No. 211.557/2021)].  

Additional Editor Comments:

1. Please provide a valid rationale for the proposed study by clearly identify and justify research questions

2. Descriptions of methods and materials in the protocol should be reported in sufficient detail

3. Discussion should be more elaborated, including implication to clinical practice

4. Please pay more attention to manuscript writing guidelines and grammar

Reviewers' comments:

Reviewer's Responses to Questions

**Comments to the Author**

1. Does the manuscript provide a valid rationale for the proposed study, with clearly identified and justified research questions?

Reviewer #1: Partly

2. Is the protocol technically sound and planned in a manner that will lead to a meaningful outcome and allow testing the stated hypotheses?

Reviewer #1: Partly

3. Is the methodology feasible and described in sufficient detail to allow the work to be replicable?

Reviewer #1: Yes

4. Have the authors described where all data underlying the findings will be made available when the study is complete?

Reviewer #1: No

5. Is the manuscript presented in an intelligible fashion and written in standard English?

Reviewer #1: Yes

6. Review Comments to the Author

You may also provide optional suggestions and comments to authors that they might find helpful in planning their study.

Reviewer #1: Comments

Line 164 , Line 77, Line 358, Line 366, Line 373. the term 'waves' could be omitted.

Line 374, the language version of Nova24h and Line 419, the amount of reimbursement is to be stated.

Line 426-427, the information provided is unclear. More information is to be provided e.g at which time point measurement and the values. In reference 53, 13.6 is referring to the age interval (months) 6.

Line -430, E= difference considered significant, δ=SD, Zα/2 =1.96 is to be denoted.

Line 430, the sample size formula presentation n=[Zα/2.δ/E]²) requires revision

Line 440, proper citation for STATA software is to be provided.

Line 447-448, the type of adipokines e.g. adiponectin and leptin are to be stated.

Preliminary results are to be omitted as the focus of the paper is describing the methodology. If the authors believe the preliminary results are important, the author may consider including them as supplementary files if allowable by the journal.

7. PLOS authors have the option to publish the peer review history of their article (what does this mean?). If published, this will include your full peer review and any attached files.

Reviewer #1: No

---

## [Author Response · Author response to Decision Letter 0]

30 Aug 2024

We revised the style according to PLOS ONE’s requirements and made the necessary corrections.

2. We note that you have selected “Clinical Trial” as your article type. PLOS ONE requires that all clinical trials are registered in an appropriate registry (the WHO list of approved registries is at https://www.who.int/clinical-trials-registry-platform/network/primary-registries" https://www.who.int/clinical-trials-registry-platform/network/primary-registries and more information on trial registration is at http://www.icmje.org/about-icmje/faqs/clinical-trials-registration/). Please state the name of the registry and the registration number (e.g. ISRCTN or ClinicalTrials.gov) in the submission data and on the title page of your manuscript. a) Please provide the complete date range for participant recruitment and follow-up in the methods section of your manuscript. b) If you have not yet registered your trial in an appropriate registry, we now require you to do so and will need confirmation of the trial registry number before we can pass your paper to the next stage of review. Please include in the Methods section of your paper your reasons for not registering this study before enrolment of participants started. Please confirm that all related trials are registered by stating: “The authors confirm that all ongoing and related trials for this drug/intervention are registered”. Please see http://journals.plos.org/plosone/s/submission-guidelines#loc-clinical-trials for our policies on clinical trials.

We did not select 'Clinical Trial' as the article type. On the submission page, we chose 'Study Protocol.' Our study is not a clinical trial but an observational cohort study. Nevertheless, the study is registered with the Brazilian Clinical Trials Registry (ReBec) under the number RBR-9hcby8c, as mentioned in the title page and methods section. The full date range for participant recruitment and follow-up has been provided and updated on page 8, line 229.

 [The project is being funded by the Research Support Program (APQ-1) of the Rio de Janeiro State Research Support Foundation (FAPERJ Proc. No. 211.557/2021)]. 

As stated in the manuscript file and now added to the cover letter:

‘The project is being funded by the Research Support Program (APQ-1) of the Rio de Janeiro State Research Support Foundation (FAPERJ Proc. No. 211.557/2021). Participant reimbursement is being provided with support from the Research Productivity Fellowship from the National Council for Scientific and Technological Development (CNPq) under the name of AEN. COM, LGR and TMT receive scholarships from Coordination for the Improvement of Higher Education Personnel (CAPES). MTNR receive scholarship from CNPq. The funders had no role in study design, data collection and analysis, decision to publish, or preparation of the manuscript.’

The data availability statement has been updated to more accurately reflect the current status of this article.

New text: ‘This article does not report any data, so the data availability policy is not applicable. Upon completion of the APPLE project, the final database containing de-identified research data will be deposited in Arca Dados (https://arcadados.fiocruz.br), the official repository of Fundação Oswaldo Cruz.’

This article does not report any data, so the data availability policy is not applicable. Upon completion of the APPLE project, the final database containing de-identified research data will be deposited in Arca Dados (https://arcadados.fiocruz.br), the official repository of Fundação Oswaldo Cruz.

This article does not report any data, so the data availability policy is not applicable. Upon completion of the APPLE project, the final database containing de-identified research data will be deposited in Arca Dados (https://arcadados.fiocruz.br), the official repository of Fundação Oswaldo Cruz.

The captions for the Supporting Information files have been added to the end of the manuscript. Additionally, we included a sentence at the end of the Methods section to provide an in-text citation for the Supporting Information.

Additional Editor Comments:

1. Please provide a valid rationale for the proposed study by clearly identify and justify research questions

A paragraph was inserted in the end of the introduction to clearly identify and justify research questions.

2. Descriptions of methods and materials in the protocol should be reported in sufficient detail

We did a careful review of the methods section, and we are convicted that all details are described.

3. Discussion should be more elaborated, including implication to clinical practice

A more detailed explanation of the implications of the results for clinical practice has been included in the Discussion section.

4. Please pay more attention to manuscript writing guidelines and grammar

Thank you, we did a careful review of writing and grammar.

Reviewer #1: Comments

Line 164 , Line 77, Line 358, Line 366, Line 373. the term 'waves' could be omitted.

Thank you for the suggestion. The word ‘waves’ was removed from the indicated sentences.

Line 374, the language version of Nova24h and Line 419, the amount of reimbursement is to be stated.

The required information has been entered in the respective lines. The reimbursement amount varies among participants. Each participant received the total amount they spent due to their participation in the project.

Line 426-427, the information provided is unclear. More information is to be provided e.g at which time point measurement and the values. In reference 53, 13.6 is referring to the age interval (months) 6.

Line -430, E= difference considered significant, δ=SD, Zα/2 =1.96 is to be denoted.

Line 430, the sample size formula presentation n=[Zα/2.δ/E]²) requires revision

Thank you for the careful revision. The formula presentation indeed needed revision, and additional information was necessary to enhance readers' understanding. We have made the required modifications.

Line 440, proper citation for STATA software is to be provided.

The proper citation was provided.

Line 447-448, the type of adipokines e.g. adiponectin and leptin are to be stated.

Thank you, the suggestion was accepted.

Preliminary results are to be omitted as the focus of the paper is describing the methodology. If the authors believe the preliminary results are important, the author may consider including them as supplementary files if allowable by the journal.

The preliminary results were removed from the manuscript.

---

## [Editor Report · Decision Letter 1]

8 Sep 2024

Plasma and breast milk adipokines in women across the first year postpartum and their association with maternal depressive symptoms and infant neurodevelopment: protocol for the APPLE prospective cohort study

PONE-D-23-34502R1

Dear Dr. Rebelo,

We’re pleased to inform you that your manuscript has been judged scientifically suitable for publication and will be formally accepted for publication once it meets all outstanding technical requirements.

Kind regards,

Annisa Dewi Nugrahani

Academic Editor

PLOS ONE
---

## [Editor Report · Acceptance letter]

26 Sep 2024

PONE-D-23-34502R1 

PLOS ONE

Dear Dr. Rebelo, 

I'm pleased to inform you that your manuscript has been deemed suitable for publication in PLOS ONE. Congratulations! Your manuscript is now being handed over to our production team.

Kind regards, 

on behalf of

Dr. Annisa Dewi Nugrahani 

Academic Editor

PLOS ONE